# Process- and product-related impurities in the ChAdOx1 nCov-19 vaccine

Lea Krutzke[1], Reinhild Rösler[2], Ellen Allmendinger[1], Tatjana Engler[1], Sebastian Wiese[2], Stefan Kochanek[1]*

[1]Department of Gene Therapy, University of Ulm, Ulm, Germany; [2]Core Unit Mass Spectrometry and Proteomics, University of Ulm, Ulm, Germany

**Abstract** ChAdOx1 nCov-19 and Ad26.COV2.S are approved vaccines inducing protective immunity against SARS-CoV-2 infection in humans by expressing the Spike protein of SARS-CoV-2. We analyzed protein content and protein composition of ChAdOx1 nCov-19 and Ad26.COV2.S by biochemical methods and by mass spectrometry. Four out of four tested lots of ChAdOx1 nCoV-19 contained significantly higher than expected levels of host cell proteins (HCPs) and of free viral proteins. The most abundant contaminating HCPs belonged to the heat-shock protein and cytoskeletal protein families. The HCP content exceeded the 400 ng specification limit per vaccine dose, as set by the European Medicines Agency (EMA) for this vaccine, by at least 25-fold and the manufacturer's batch-release data in some of the lots by several hundred-fold. In contrast, three tested lots of the Ad26.COV2.S vaccine contained only very low amounts of HCPs. As shown for Ad26.COV2.S production of clinical grade adenovirus vaccines of high purity is feasible at an industrial scale. Correspondingly, purification procedures of the ChAdOx1 nCov-19 vaccine should be modified to remove protein impurities as good as possible. Our data also indicate that standard quality assays, as they are used in the manufacturing of proteins, have to be adapted for vectored vaccines.

## Editor's evaluation

This research shows that a commonly used commercial vaccine for COVID-19 harbors contaminating proteins derived from the human cell line in which it is produced. The health significance of these contaminants (if any) remains unknown. This paper is important because lot purity and processing of vaccines is rarely scrutinized in the scientific realm, and instead is typically analyzed only by the companies themselves.

*For correspondence:
stefan.kochanek@uni-ulm.de

Competing interest: The authors declare that no competing interests exist.

## Introduction

Beside mRNA vaccines, adenoviral vector-based vaccines have turned out to be an essential mainstay of the vaccination campaign against COVID-19. The AstraZeneca COVID-19 vaccine (AZD1222, Vaxzevria, ChAdOx1 nCov-19 – short ChAdOx1) is based on Chimpanzee Adenovirus Y25 (*Dicks et al., 2012*) . The Johnson & Johnson Ad26.COV2.S vaccine is based on Human Adenovirus type 26 (HAdV-D26) (*Bos et al., 2020*). Both vaccines express the full-length Spike protein of SARS-CoV-2. The ability of adenovirus-based vaccines to induce potent humoral and cellular immune responses and their overall safety record had made them strong candidates to successfully fight the COVID-19 pandemic. This has turned out true, with billions of adenovirus-based vaccine doses administered to vaccinees and with very low rates of serious adverse events observed. In the past, and before the COVID-19 pandemic, adenovirus-based vaccines have not been produced even close at the scale required to supply billions of doses. Thus, for all vaccines including for adenovirus-based vaccines scale-up of production processes in a very short period of time has been a significant challenge.

## Results

### Protein impurities detected in the AstraZeneca ChAdOx1 nCov-19 vaccine

We initially analyzed three different lots of ChAdOx1 by sodium dodecyl sulfate–polyacrylamide gel electrophoresis (SDS–PAGE) followed by silver staining, comparing the staining pattern of the proteins with those of HAdV-C5-EGFP, an HAdV-C5-based adenovirus vector expressing EGFP, purified by CsCl ultracentrifugation. The vaccine is produced in human T-REx-293 cells to prevent expression of the SARS-CoV-2 spike protein during vector production. The vaccine is then purified by a combination of filtration steps and anion-exchange chromatography (*European-Medicines-Agency, 2021a*; *Fedosyuk et al., 2019*). Previously, it has been shown, that simian adenovirus vectors including ChAdOx1 can be purified at high yield and purity using such technology for purification (*Fedosyuk et al., 2019*). To meet global needs and to enable commercial manufacturing of clinical grade material at a very large scale, some modifications and simplifications, respectively, had been introduced both into the upstream and downstream processes, as has been described in detail (*Joe et al., 2022*). Although in our experiment the same number of viral particles was loaded, the staining pattern of ChAdOx1 looked very different, when compared to the adenoviral vector control (*Figure 1*).

While staining of HAdV-C5-EGFP proteins resulted in the expected band pattern, representing distinct major capsid proteins of HAdV-C5 including Hexon, Penton Base, IIIa, and Fiber, protein staining of three different lots of ChAdOx1 showed many more bands than could be explained by proteins from viral particles. Additionally, despite loading the same number of particles, band intensities varied between the ChAdOx1 lots.

We also analyzed the ChAdOx1 nCoV-19 lots at the DNA level by quantitative polymerase chain reaction (PCR). Results confirmed the exclusive presence of viral DNA, while genomic DNA of the host cell was not detected (data not shown).

### Detection of substantial amounts of host cell proteins in the vaccine

To determine the protein composition of the vaccine, we performed mass spectrometry (MS) analyses from either tryptic in-solution digest directly from the vaccine (*Figure 2*) or from tryptic in-gel-digest after SDS–PAGE, with similar results (*Figure 2—figure supplement 1* and *Figure 2—source data 1*).

Based on intensity comparisons of liquid chromatography/mass spectrometry (LC/MS) signals, we estimate that in lot ABV5811 about 70% of the detected protein content was of human and only 30% of virus origin. In lots ABV4678 and ABV7764 approximately 50% of detected proteins were of human origin (*Figure 2A*).

Beside the expected structural viral proteins forming the virion (Hexon, Penton base, IIIa, Fiber, V, VI, VII, VIII, IX, and others) also several nonstructural viral proteins were detected at high abundancy, although they are not part of the mature viral particle (*Figure 2B,C* and *Figure 2—figure supplement 2*). To the detected nonstructural viral proteins belong, for example, the 100K protein, a multifunctional scaffolding protein involved in the trimerization of Hexon, and the DNA-binding protein, which plays an essential role in the replication of the viral genome during infection. In lot ABV5811 nonstructural proteins represented 30% of detected adenoviral proteins (*Figure 2B*). Since in the assembly process during virus propagation only a part of the available viral capsid proteins is used for particle formation, we assume that in the vaccine product also significant amounts of structural proteins were present as monomers, oligomers, or incomplete viral capsid assemblies. Not being part of the mature viral particle, both nonstructural viral proteins and nonencapsidated structural proteins are referred to as product-related impurities.

Peptides from more than 1000 different human proteins, derived from the human production cell line, were detected (*Figure 2—source data 1*). They were derived from different cellular compartments including cytoplasm, nucleus, endoplasmic reticulum, Golgi apparatus, and others. Relative amounts of human versus viral proteins were variable between lots (*Figure 2A*), as was already assumed based on the silver-stained gels (*Figure 1*). Among the human proteins found in the vaccine and beside several cytoskeletal proteins including Vimentin, Tubulin, Actin, and Actinin, the group of heat-shock proteins (HSPs) and chaperones stood out in abundancy. Among the top abundant proteins, HSP 90-beta and HSP 90-alpha as cytosolic HSPs (9.5% and 4.3% of the total proteins, respectively) and three chaperones of the endoplasmic reticulum (transitional endoplasmic reticulum ATPase, Endoplasmin, and Calreticulin) were present (*Figure 2C* and *Figure 2—figure supplement 2*).

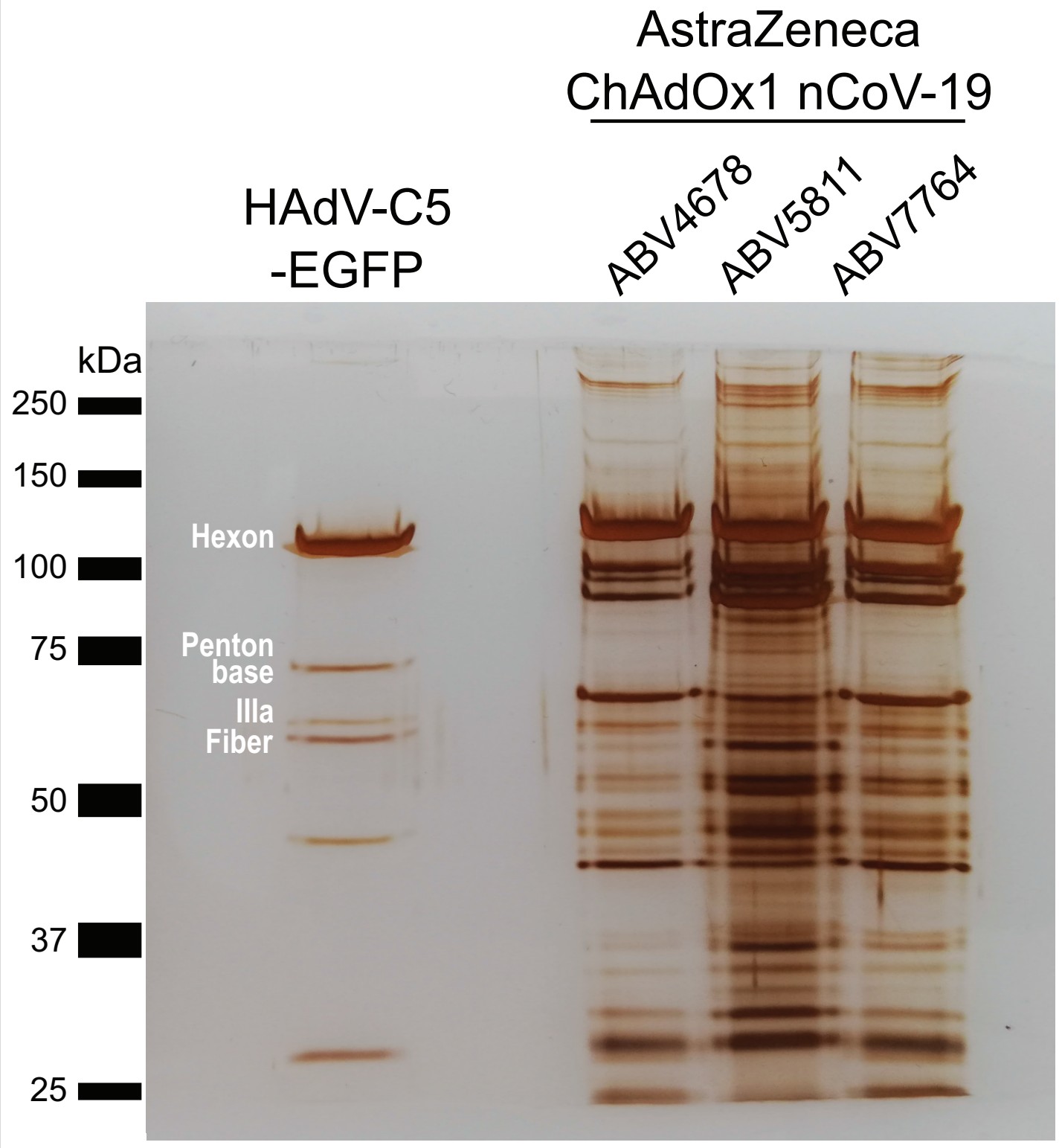

**Figure 1.** Protein staining of HAdV-C5-EGFP and three ChAdOx1 nCoV-19 vaccine lots. 3 × 10⁹ adenoviral vector particles were separated by sodium dodecyl sulfate–polyacrylamide gel electrophoresis (SDS–PAGE) under denaturing and reducing conditions. Proteins were visualized by silver staining. Known HAdV-C5 proteins are labeled. Three different vaccine lots (ABV4678, ABV5811, and ABV7764) of ChAdOx1, produced by the manufacturer, were analyzed.

The online version of this article includes the following source data for figure 1:

*Figure 1 continued*
**Source data 1.** Original file of the full raw unedited gel: Protein staining of HAdV-C5-EGFP and three ChAdOx1 nCov-19 vaccine lots.
**Source data 2.** Uncropped gel with the relevant bands labeled: Protein staining of HAdV-C5-EGFP and three ChAdOx1 nCov-19 vaccine lots.

Proteins from Bos taurus from fetal calf serum used for growth of T-REx-293 producer cells, Spike protein of SARS-CoV-2 and T-Rex-293 cell-derived proteins (E1B from HAdV-C5, Tet-Repressor) were detected at low or negligible levels (*Figure 2—source data 1*).

## Lack of host cell proteins in the Ad26.COV2.S vaccine

A second approved adenoviral vaccine, based on HAdV-D26, is also purified using filtration and chromatography steps (*European-Medicines-Agency, 2021b*). To our knowledge, more detailed information on the purification process is not publicly available. We compared again a CsCl-purified HAdV-C5-EGFP adenovirus vector with three lots of Ad26.COV2.S (lots 21C10-01, XD955, XE395) and a further (fourth) lot of ChAdOx1 (ABV9317) (*Figure 3*) and found very little protein contamination in Ad26.COV2.S compared to ChAdOx1 (*Figure 3A-C* and *Figure 3—source data 1*).

Mass spectrometry confirmed that more than 93% of proteins detected in HAdV-C5-EGFP were of HAdV-C5 origin (*Figure 3—figure supplement 1*). Furthermore, 98% of proteins detected in Ad26. COV2.S (lot XD955) were of HAdV-D26 and less than 1% of human origin. In contrast, in lot ABV9317 of ChAdOx1 70% of proteins were of human and only 30% of adenovirus origin (*Figure 3B,C* and *Figure 3—figure supplement 2*). Determination of structural and nonstructural viral proteins revealed viral structural to nonstructural protein ratios of 92% versus 8% and of 70% versus 30% in Ad26. COV2.S compared to ChAdOx1, respectively (*Figure 3D*). The top human host cell proteins (HCPs) in Ad26.COV2.S, although present only at very low abundancy, were histones and ribosomal proteins, rather than the HSPs and cytoskeletal proteins found in ChAdOx1 (*Figure 3—source data 1*).

## Quantification of HCPs present in different ChAdOx1 vaccine lots

According to European Medicines Agency (EMA) quality specifications for ChAdOx1, 400 ng is the maximally allowable HCP content per single dose ($5 \times 10^{10}$ vector particles). Since the quantification of total protein amounts in inhomogeneous protein mixtures is inherently afflicted with uncertainty due to differences in biophysical properties of the individual proteins, we used three standard methods to determine the HCP content in the tested ChAdOx1 lots (*Table 1*).

We took into account the 12.8 µg protein amount contained in $5 \times 10^{10}$ vector particles, that can be calculated based on the known 150 MDa molecular weight of adenovirus particles (*Sweeney and Hennessey, 2002*). As expected, and as apparent from *Table 1*, total protein contents varied with the method used for analysis. However, independent of the method used and as already expected from inspection of *Figure 3A*, protein amounts contained in the vaccine significantly exceeded the 400 ng specification limit by a factor of 25 or higher, and the manufacturer's batch-release data by an even much larger (several hundred-fold) extent.

## In vivo study to analyze potential effects of process- and product-related impurities on anti-Spike immune responses

To test whether the process- and product-related impurities in the ChAdOx1 nCov-19 vaccine might influence immune responses against the encoded SARS-CoV-2 Spike protein upon vaccination, we performed an in vivo immunization experiment in mice. We amplified ChAdOx1 nCov-19 in HEK293T cells and purified this vector to more than 90% purity (renamed UUlm ChAdOx1 nCov-19) by standard CsCl density gradient ultracentrifugation (*Figure 4A*, *Figure 4—figure supplement 1*, *Figure 4—source data 1*). BALB/c mice were immunized by intramuscular injection of $1 \times 10^9$ vector particles of AstraZeneca ChAdOx1 nCov-19 (ABV9317) or UUlm ChAdOx1 nCov-19. Two weeks later, spike protein-specific T-cell responses (*Figure 4B*) and spike protein-specific total antibody titers (*Figure 4C*) were determined.

T-cell responses (against one tested T-cell epitope) were 1.3-fold lower after vaccination using UUlm ChAdOx1 nCov-19 compared to vaccination using AstraZeneca ChAdOx1 nCov-19. Similarly, total antibody titers were also 1.3-fold lower after vaccination using UUlm ChAdOx1 nCov-19 compared to vaccination using AstraZeneca ChAdOx1 nCov-19. However, differences between the two groups

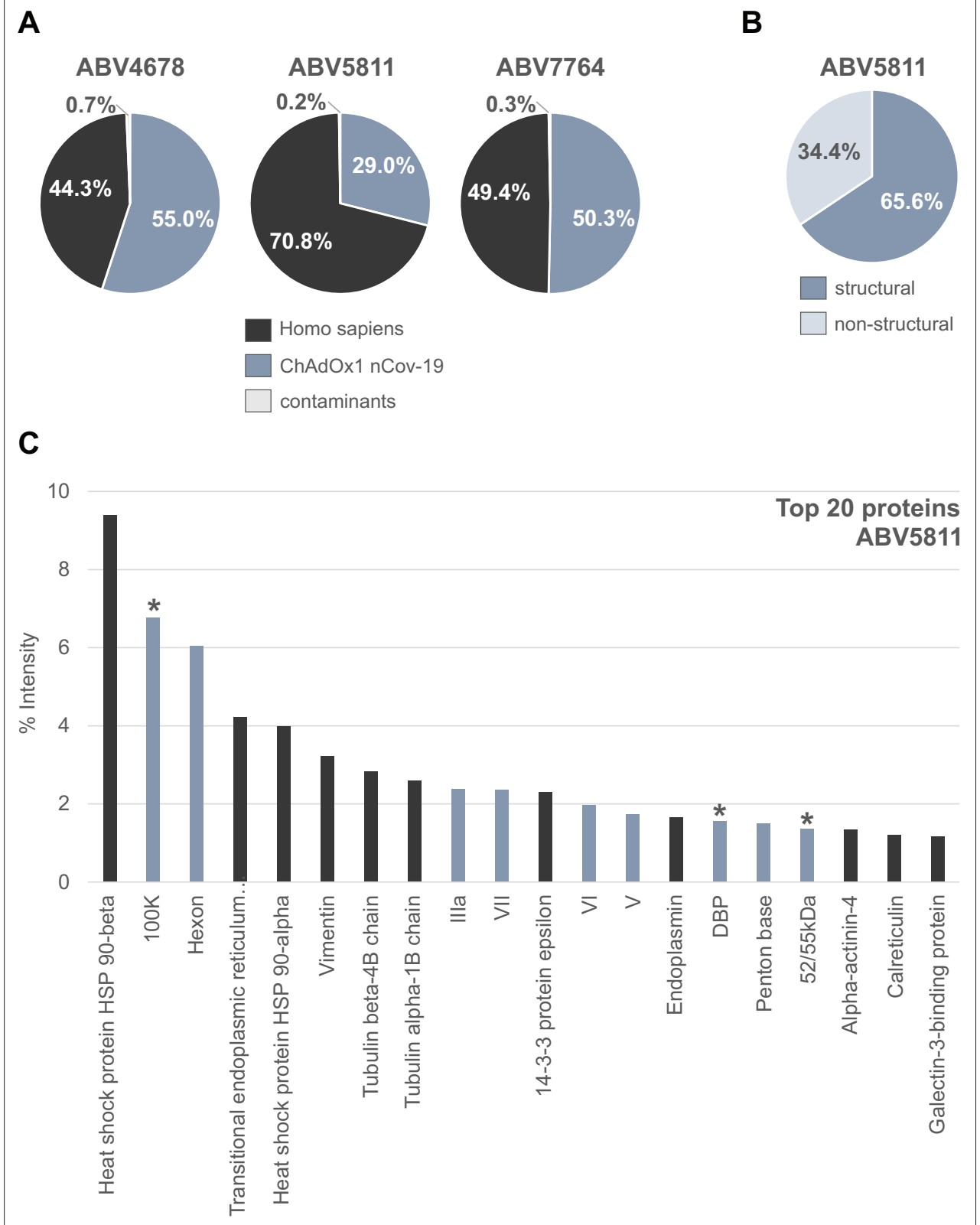

**Figure 2.** Distribution of proteins in three ChAdOx1 nCov-19 vaccine lots. The protein composition of three lots of ChAdOx1 (ABV4678, ABV5811, and ABV7764) was analyzed by mass spectrometry following in-solution protein digest. Spectral data were aligned via search engine with human and viral databases (*Figure 2—source data 1*). (**A**) Percentage of total intensities associated with proteins from the respective organism were subsequently summed. (**B**) Percentage of structural and nonstructural adenoviral proteins detected in ChAdOx1 nCov-19 vaccine lot ABV5811. (**C**) Intensity

*Figure 2 continued on next page*

*Figure 2 continued*

distribution of the top 20 proteins of the ChAdOx1 nCov-19 vaccine lot ABV5811 detected. Proteins that originate from *Homo sapiens* are depicted in black; proteins that originate from ChAdOx1 are depicted in blue-gray; *nonstructural adenoviral proteins.

The online version of this article includes the following source data and figure supplement(s) for figure 2:

**Source data 1.** List of proteins detected in three ChAdOx1 nCov-19 vaccine lots by mass spectrometry.

**Figure supplement 1.** Distribution of proteins of the ChAdOx1 nCov-19 vaccine following PAGE and in-gel digests.

**Figure supplement 2.** Intensity distribution of the top 20 proteins in ChAdOx1 nCov-19 vaccine lots.

were not statistically significant in both parameters, indicating that in this limited study performed in mice there was no indication for obvious interference in induced immune responses related to the HCPs in the original AstraZeneca ChAdOx1 nCov-19 vaccine.

## Detection of ATPase activity in the ChAdOx1 nCov-19 vaccine

Since HSPs and chaperones, found at substantial levels in the ChAdOx1 nCov-19 vaccine are equipped with ATPase activity and because ADP is a strong activator of platelets, we analyzed, whether ATPase activity was detectable in the ChAdOx1 nCov-19 vaccine. ChAdOx1 nCov-19 (lot ABV9317) was incubated with ATP and the hydrolysis of ATP to ADP was determined using a commercial assay. Results confirmed the presence of ATPase activity in the vaccine (*Figure 5A,B*).

## Discussion

When analyzing the protein composition of ChAdOx1, a vaccine with an overall good safety record and with high efficacy in preventing or ameliorating COVID-19, we found that the HCP content in ChAdOx1 exceeded the 400 ng/dose specification limit, as set by the EMA, by at least 25-fold in all tested lots, in some of the lots substantially more (*Table 1*). Compared to the manufacturer's batch-release data the difference between actual and assumed HCP content, respectively, was even several hundred-fold in some of the lots. The substantial lot-to-lot variation in HCP content in a total of four tested samples (*Figure 1* and *Table 1*) also indicated a lack of robustness of the process used for purification of the vaccine, a process, which recently has been described in detail (*Joe et al., 2022*). In contrast, in three tested lots of the Ad26.COV2.S vaccine, contamination with HCPs was negligible, confirming that excellent purification of clinical grade adenovirus vaccines is possible at an industrial scale.

One can assume that the vast majority of the more than 1000 different proteins detected, most of which present at low abundancy, will not cause any adverse effects. We note that the most abundant proteins do not belong to the group of 'high risk' HCPs such as cytokines, proteases, and lipases, as they have been described in the context of protein therapeutics (*Jones et al., 2021*). Whether or not some of the detected human and/or viral protein contaminants might enhance early clinical vaccine reactions such as flu-like symptoms, very often observed within 1 or 2 days after intramuscular injection of vaccines, is a question that cannot be easily addressed.

There is a possibility that some of the proteins, in particular those with higher abundance, might be more than inert bystanders. For example, extracellular HSPs are known to modulate innate and adaptive immune responses, can exacerbate preexisting inflammatory condition, have been associated with autoimmunity and can even become targets auf autoimmune responses themselves (*Binder, 2014*; *Routsias and Tzioufas, 2006*; *Tamura et al., 2016*). They very efficiently initiate specific immune responses by receptor-mediated uptake of HSP–peptide complexes in antigen-presenting cells (APCs), mainly via CD91 and scavenger receptors (*Binder, 2014*; *Binder et al., 2000*). Since the HSPs present in the vaccine are derived from T-REx-293 cells, in principle they could mediate the transfer to APCs of peptides derived from the 293 cell source, of autologous peptides from vaccinated individuals and also of viral proteins. Like many viruses, adenovirus has been reported to induce HSPs during amplification in production cells (*Santoro et al., 2010*), likely to accommodate for a need in help by HSPs in the folding and production of large amounts of structural and nonstructural viral proteins and in virion assembly. Thus, virus infection-mediated induction of HSPs could explain the abundancy of HSPs in the vaccine product in case of insufficient purification.

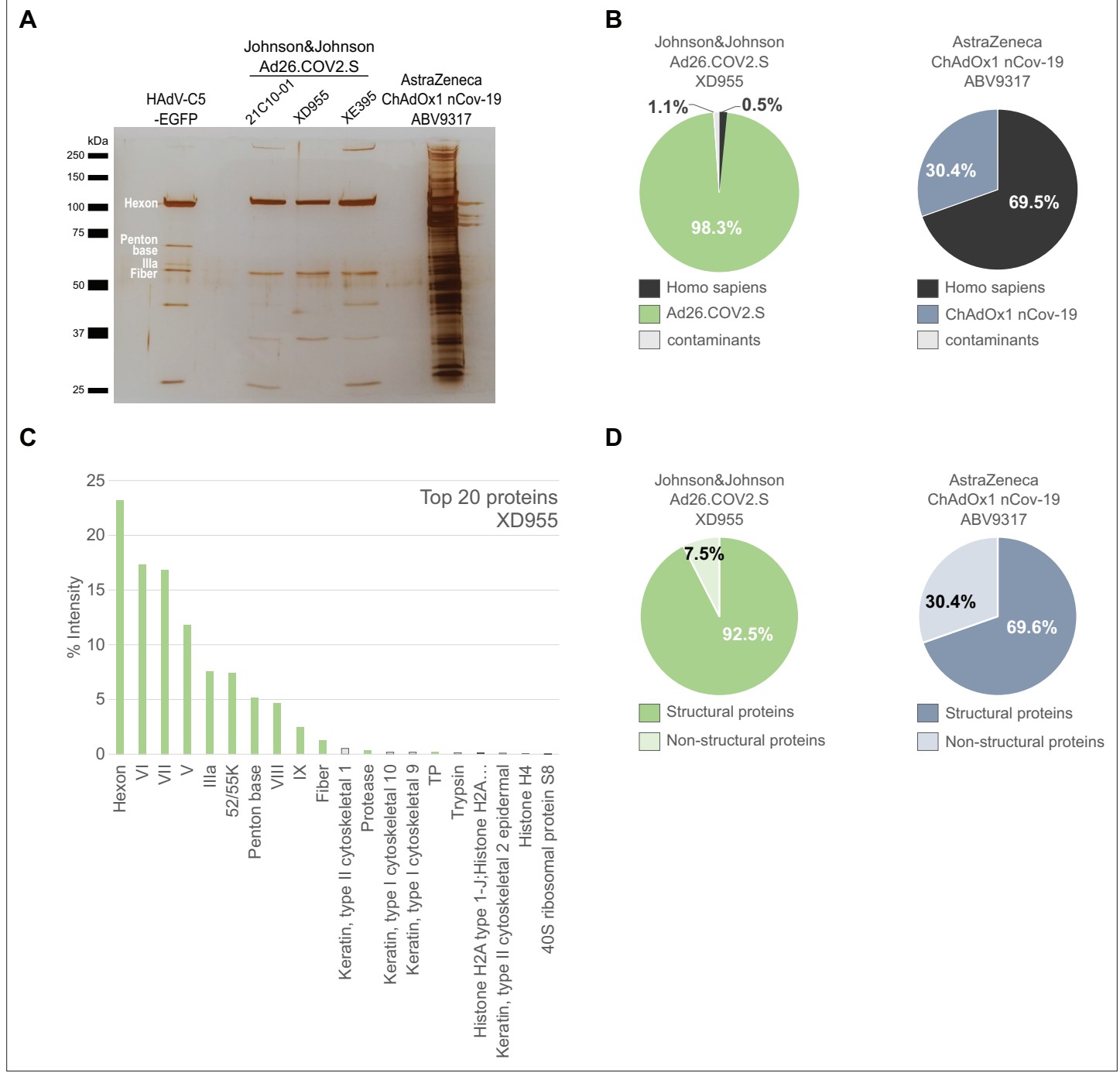

**Figure 3.** Comparison of Ad26.COV2.S and ChAdOx1 nCov-19 vaccines by biochemical and proteomic analysis.
 (**A**) Proteins corresponding to each 3 × 10⁹ vector particles of three lots of Ad26.COV2.S (21C10-01, XD955, and XE395), one lot of ChAdOx1 (ABV9317) and CsCl-purified HAdV-C5-EGFP (control), as indicated, were separated by sodium dodecyl sulfate–polyacrylamide gel electrophoresis (SDS–PAGE) performed under denaturing and reducing conditions. Proteins were visualized by silver staining. Known HAdV-C5 proteins are indicated. Note that on SDS–PAGE the stained protein band corresponding to Penton Base of HAdV-C5 (571 amino acids, predicted molecular weight [MW]: 63.4 kDa) has a slower migration behavior than expected, likely due to a flexible loop that is structurally disordered in HAdV-C5 (*Flatt et al., 2013*) and known to be hypervariable in different adenovirus types (*Zubieta et al., 2005*). Compared to HAdV-C5, in Chimpanzee Adenovirus Y25 (532 amino acids, MW: 532 kDa) and in HAdV-D26 (519 amino acids, MW: 58.6) this loop is reduced in size and, according to the PONDR-Fit online tool (*Xue et al., 2010*), less disordered (data not shown), explaining, why in Ad26.COV2.S and in ChAdOx1 the Penton Base-corresponding signals likely overlap with those of the smaller viral proteins IIIa or fiber. (**B–D**) Ad26.COV2.S (lot XD955) and ChAdOx1 nCov-19 (lot ABV9317) were analyzed by mass spectrometry following in-solution protein digest. Spectral data were aligned via search engine with human and viral proteins (*Figure 3—source data 1*). Human proteins are

Figure 3 continued

depicted in black; proteins that originate from ChAdOx1 are depicted in blue-gray; proteins that originate from Ad26.COV2.S are depicted in green. (**B**) Percentage of total intensities of proteins from the respective organism detected in the Ad26.COV2.S (lot XD955) and ChAdOx1 nCov-19 (lot ABV9317) vaccines. (**C**) Intensity distribution of the top 20 proteins of the Ad26.COV2.S vaccine (lot XD955). (**D**) Percentage of total intensities of structural and nonstructural adenoviral proteins detected in Ad26.COV2.S (lot XD955) and ChAdOx1 nCov-19 (lot ABV9317).

The online version of this article includes the following source data and figure supplement(s) for figure 3:

**Source data 1.** List of proteins detected by mass spectrometry in three Ad26.COV2.S vaccine lots, ChAdOx1 nCov-19 vaccine lot ABV9317, and HAdV-C5-EGFP.

**Source data 2.** Original file of the full raw unedited gel: Comparison of Ad26.COV2.S and ChAdOx1 nCov-19 vaccines.

**Source data 3.** Uncropped gel with the relevant bands labeled: Protein staining of HAdV-C5-EGFP, three Johnson&Johnson Ad25-COV2.S vaccine lots and one AstraZeneca ChAdOx1 nCov-19 vaccine lot.

**Figure supplement 1.** Distribution of proteins of the HAdV-C5-EGFP vector following in-solution protein digest.

**Figure supplement 2.** Distribution of proteins of the Johnson & Johnson Ad26.COV2.S vaccine lots 21C10-01 and lot XE395 following in-solution protein digest.

Because HSPs are endowed with ATPase activity, we searched for and detected ATPase activity in ChAdOx1 (**Figure 5**)**,** raising the possibility that local ADP generation in the ATP-rich skeletal muscle tissue after injection might contribute to activation of platelets following ADP receptor binding.

Shortly after receiving conditional marketing authorization in Europe, cases of thrombotic thrombocytopenia were observed, an unusual and, if untreated, often fatal syndrome of thrombosis of the cerebral venous sinuses and other large vessels together with low platelet counts. Occurring between 5 and 28 days after vaccination (**Othman et al., 2021**), this syndrome, now coined vaccine-induced immune thrombotic thrombocytopenia (VITT) or thrombosis with thrombocytopenia syndrome, and related to autoimmune heparin-induced thrombocytopenia (**Greinacher et al., 2017**), was found to be linked to vaccination with ChAdOx1 (**Paul-Ehrlich-Institut, 2021**), and has been described to be mediated by platelet-activating antibodies against platelet factor 4 (PF4) (**Greinacher et al., 2021a** ; **Greinacher et al., 2021b**; **Pavord et al., 2021**; **Schultz et al., 2021**; **Scully et al., 2021**; **Tiede et al., 2021**; **Wolf et al., 2021**). Also Ad26.COV2.S has been linked to VITT (**Muir et al., 2021**; **Paul-Ehrlich-Institut, 2021**), however at a lower frequency (**Paul-Ehrlich-Institut, 2022**). It has recently been speculated that impurities in ChAdOx1 could be causally involved in the pathogenesis of VITT (**Greinacher et al., 2021a**). Since VITT has been found associated with both ChAdOx1 and Ad26.COV2.S, at a lower frequency in the latter (**Paul-Ehrlich-Institut, 2021**), and because, according to our data, ChAdOx1 but not Ad26.COV2.S contained high amounts of human and free viral proteins, our findings support the notion that the protein impurities present in ChAdOx1 at least are not the primary trigger for the induction of VITT. Incidentally, PF4 itself was not found in our analyses as a contaminant. Whether the impurities might increase the likelihood and frequency of its occurrence in case of ChAdOx1, for example by serving as an inflammatory or immunological cofactor remains a possibility. Interestingly, epidemiologic studies have indicated that RNA vaccines are associated with thromboembolic events but not with VITT (**Paul-Ehrlich-Institut, 2021**). We speculate that differences in duration and/or levels

**Table 1.** Amount of protein impurities per vaccine dose in the ChAdOx1 nCoV-19 vaccine. European Medicines Agency (EMA) quality specifications of accepted host cell protein (HCP) amounts in the ChAdOx1 nCoV-19 vaccine and batch-release data by the manufacturer, obtained through the German information freedom act. Total protein content per dose as determined by three different methods of lots ABV4678, ABV5811, ABV7764, and ABV9317 after subtraction of 12.8 µg virus protein contained in $5 \times 10^{10}$ vector particles *Sweeney and Hennessey, 2002*. n = 3.

| ChAdOx1 nCov-19 lot # | EMA specification [µg HCP/dose] | Batch-release data manufacturer [µg HCP/dose] | NanoDrop absorbance 280 **nm** [µg HCP/dose] | Bradford assay [µg HCP/dose] | ImageJ analysis [µg HCP/dose] |
|---|---|---|---|---|---|
| ABV4678 | ≤0.4 | 0.05 | 39.03 | 12.36 | 9.85 |
| ABV5811 | ≤0.4 | 0.3 | 114.53 | 24.78 | 24.9 |
| ABV7764 | ≤0.4 | 0.0489 | 74.37 | 29.66 | 12.81 |
| ABV9317 | ≤0.4 | n.a. | 154.53 | 42.48 | 73.16 |

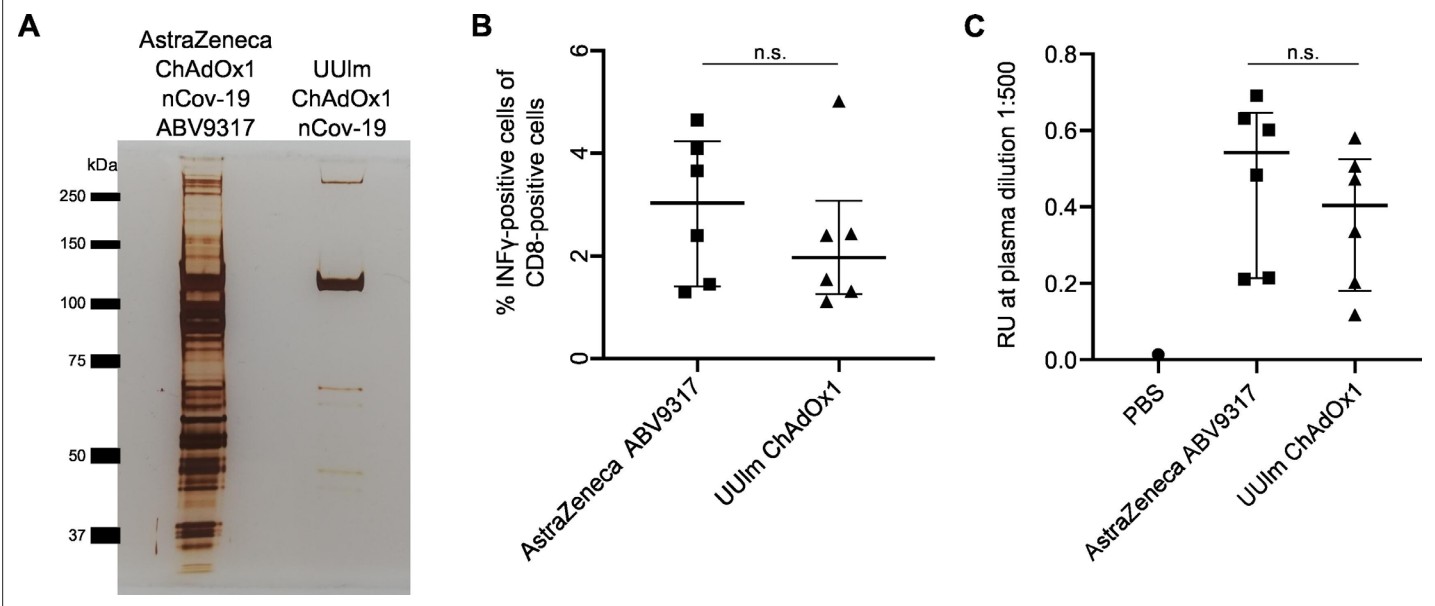

**Figure 4.** Analysis of effects of impurities in the ChAdOx1 nCoV-19 vaccine on anti-Spike immune responses in vivo. BALB/c mice were injected intramuscularly with phosphate-buffered saline (PBS) or $1 \times 10^9$ vector particles of AstraZeneca ChAdOx1 nCoV-19 vaccine (ABV9317) or UUlm ChAdOx1 nCov-19 dissolved in PBS. Fourteen days later, mice were sacrificed and plasma and spleen samples collected. $n$ = 6/group. (**A**) $3 \times 10^9$ adenoviral vector particles of AstraZeneca ChAdOx1 nCoV-19 vaccine (ABV9317) or UUlm ChAdOx1 were separated by sodium dodecyl sulfate–polyacrylamide gel electrophoresis (SDS–PAGE) under denaturing and reducing conditions. Proteins were visualized by silver staining. (**B**) SARS-CoV-2 spike protein-directed T-cell responses were measured by stimulation of isolated T cells with spike peptide-loaded dimer CoVK[d-268] GYLQPRTFL in duplicates and subsequent intracellular staining of INFγ. Results are given as mean % INFγ-positive cells of CD8-positive cells of single animals. (**C**) SARS-CoV-2 spike protein-specific antibody titers were determined by enzyme-linked immunosorbent assay (ELISA). Purified full-length spike protein was coated and serial plasma dilutions were added in duplicates. Results are given as mean response units (RU) of single animals at representative plasma dilutions of 1:500. n.s.: statistically not significant.

The online version of this article includes the following source data and figure supplement(s) for figure 4:

**Source data 1.** List of proteins detected by mass spectrometry in UUlm ChAdOx1 nCov-19.

**Source data 2.** Original file of the full raw unedited gel: Protein staining of ChAdOx1 nCoV-19 vaccine lot and UUlm ChAdOx1.

**Source data 3.** Uncropped gel with the relevant bands labeled: Protein staining of ChAdOx1 nCoV-19 vaccine lot and UUlm ChAdOx1.

**Figure supplement 1.** Distribution of proteins of the UUlm ChAdOx1 nCov-19 vector following in-solution protein digest.

of Spike protein expression, as to be expected in adenoviral vectors versus mRNA vaccines, might also be a relevant factor in the pathogenesis of VITT, as well as the secretion of the Spike protein due to aberrant splicing events, as recently observed with adenovirus-based vaccines, more so with ChAdOx1 than with Ad26.COV2.S (*Kowarz et al., 2022*). However, it is important to stress that the frequency of severe thromboembolic events occurring after natural infection with SARS-CoV-2 is much higher than after vaccination with any of the vaccines in use including ChAdOx1 and Ad26.COV2.S (*Lau et al., 2021*; *Nopp et al., 2020*).

We also considered that intramuscular injection of contaminating free viral or nonviral proteins might influence the quality of the immune response and potentially affect activity and efficacy of the vaccine. In a pilot in vivo immunization experiment performed in mice, anti-Spike antibody and T-cell responses were not significantly altered following immunization with ChAdOx1 compared to an identical vector without HCP contamination (*Figure 4*), suggesting that the protein impurities did not interfere with the induction of immune responses against the Spike antigen. However, we acknowledge that our study was limited by testing recognition of only a single T-cell epitope.

In the biopharmaceutical industry, the removal of HCPs from the biological product is a critical quality attribute, since residual HCPs pose a risk to patient safety (*Vanderlaan et al., 2018*; *Wang et al., 2009*; *Zhu-Shimoni et al., 2014*). The main risks generally discussed relate either to the biological activity of the contaminant, to effects of the impurities enhancing immune responses against the therapeutic protein (antidrug antibodies) or to inducing antibodies against the impurities (*Vanderlaan*

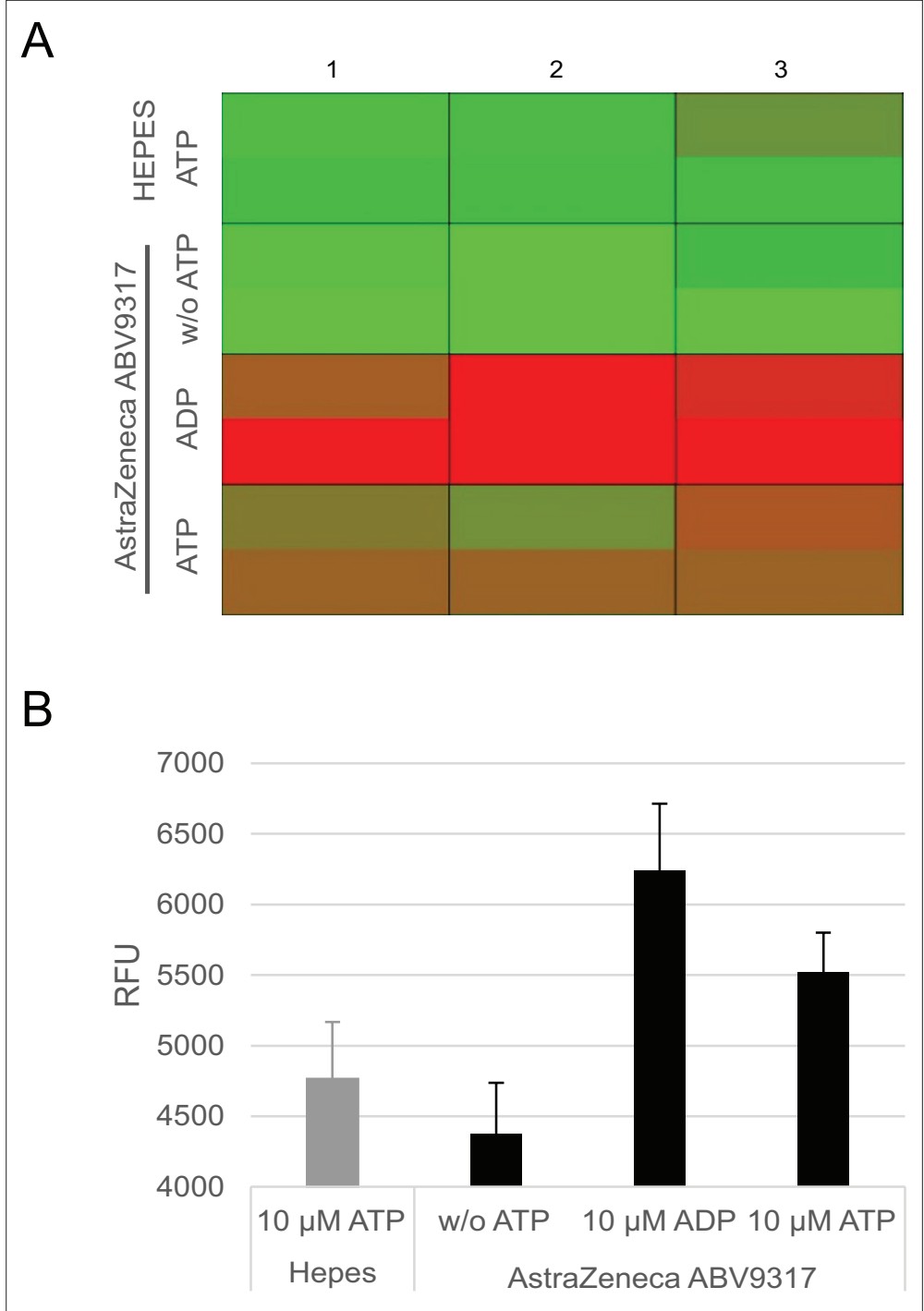

**Figure 5.** Analysis of ATPase activity in the ChAdOx1 nCoV-19 vaccine. Conversion of ATP into ADP by ATPase activity of ChAdOx1 nCoV-19 vaccine lot ABV9317 was analyzed using a fluorescence-based commercially available ADP$^2$ FI Assay kit, showing increasing fluorescence intensity signals with increasing amounts of ADP. Negative controls: 4-[2-hydroxyethyl]-1-piperazineethanesulfonic acid (HEPES) instead of ChAdOx1 nCov-19, no addition of ATP; Positive control: addition of ADP instead of ATP. (**A**) Heatmap of fluorescence intensity of Alexa594 fluorophore. (**B**) Quantification of measured fluorescence intensity. RFU = response fluorescence units. $n$ = 3.

*et al., 2018*). These concerns are relevant as they have been observed clinically. Overall, problems with HCPs are rare events and classical protein biopharmaceuticals have an excellent safety record (*de Zafra et al., 2015*). In cases, in which problems have been observed, therapeutic proteins have been produced in heterologous (i.e., nonhuman) production systems such as *E. coli*, yeast, and Chinese Hamster Ovary (CHO) cell lines (*de Zafra et al., 2015*). For example, antidrug antibodies were generated in patients against human growth hormone (*Vanderlaan et al., 2018*), produced in *E. coli*, apparently due to a contaminating protein that had been missed by an unsuitable HCP assay. Another case, more closely related to the topic of this report, is bovine neonatal pancytopenia, a severe vaccine-induce alloimmune disease in young calves that is caused by alloreactive maternal antibodies present in the colostrum of cows that had been vaccinated against bovine viral diarrhea (*Kasonta et al., 2018*). Bioprocess impurities in the vaccine, originating from the bovine i.e. autologous, production cell line, used for virus propagation and vaccine production, induced these alloreactive antibodies present in the colostrum of vaccinated cows that were directed to polymorphic bovine leukocyte antigen I (BoLA class I) alleles and became symptomatic in the offspring (*Kasonta et al., 2018*). In principle, for biopharmaceuticals used in humans one can expect lower immune-related risks linked to HCP impurities when they are produced in human cell lines (*de Zafra et al., 2015*).

Guidance documents by the US Pharmacopeia (USP) and the European Pharmacopeia emphasize that HCP impurities should be as low as technically feasible (*Vanderlaan et al., 2018*). For 'classical' biopharmaceuticals such as antibodies the allowable HCP content generally is in a range of 1–100 parts per million (1–100 ppm), corresponding to 1–100 ng HCP in 1 mg product (*Wang et al., 2009*; *Vanderlaan et al., 2018*). The HCP content in more complex biotherapeutics such as viral vectors will be higher than in protein therapeutics, since production processes are more demanding. Therefore, HCP specifications are individually determined by the regulatory bodies. One of the likely reasons, why in the case of ChAdOx1 the specification limit of 400 ng HCP content per dose was largely exceeded (*Table 1*), has to do with the assays used for HCP determination in the purified product. Standard assays to monitor removal of HCPs during production of biopharmaceuticals are enzyme-linked immunosorbent assays (ELISAs) (*Wang et al., 2009*; *Vanderlaan et al., 2018*; *European Directorate for the Quality of Medicine and Health Care, 2020*), based on polyclonal antibodies isolated from larger animals, preferentially goat, sheep, and rabbit, after immunization with cell lysates or supernatants from the producer cell. For production of secreted recombinant proteins, such assays are a 'workhorse' in quality control (*Vanderlaan et al., 2018*), routinely used and together with other methods to assure and document the absence of HCPs in the final product. The apparent lack of detection by the manufacturer of HSPs and cytoskeletal proteins in ChAdOx1 with an ELISA could be explained by the close to 100% homology at the protein level of HSPs and cytoskeletal proteins between human and sheep or goat, so that in immunized animals no or low-level antibodies against these protein classes were generated. Thereby, these proteins could be missed, if quality control was solely based on such a polyclonal ELISA, in particular, if it was not complemented by orthogonal analytical methods such as staining of protein gels, capillary electrophoresis, or mass spectrometry.

With the many different contaminant proteins detected in the ChAdOx1 nCov-19 vaccine the question imposes itself, whether or not (some of) the impurities might have long-term immune-related side effects in some of the vaccinees. In order to reduce potential risks as good as possible it will be necessary to reduce the level of contamination with process- and product-related impurities by an improved purification process. The establishment of robust assays for detection of HCPs can be complicated and time consuming (*Wang et al., 2009*), in particular if processes for new and complex biopharmaceuticals have to be developed, time that maybe is not always available in times of a pandemic. However, the here reported identification of specific process- and product-related impurities in ChAdOx1 should guide and accelerate the next steps to improve purity and quality of this and similar vaccines in the future.

# Materials and methods

**Key resources table**

| Reagent type (species) or resource | Designation | Source or reference | Identifiers | Additional information |
|---|---|---|---|---|
| Other | Covid-19 vaccine ChAdOx1 nCov-19 | AstraZeneca, distributed by the Pharmacy of the University Hospital Ulm, Germany | Lot: ABV4678 ABV5811 ABV7764 ABV9317 | Commercial vaccine |
| Other | Covid-19 vaccine Ad26-COV2.S | Johnson & Johnson, distributed by the Pharmacy of the University Hospital Ulm, Germany | Lot: XD955 21C10-01 XE395 | Commercial vaccine |
| Biological sample (Human Adenovirus type 5) | HAdV-C5-EGFP (ΔE1; CMV-eGFP) | GenBank: AY339865.1 (Δ nt 441–3522) | | +CMV promoter-controlled eGFP expression cassette |
| Cell line (*Homo sapiens*) | N52.E6 | *Schiedner et al., 2000*, Human Gene Therapy 11, 2105–2116, 2000 | | |
| Cell line (*Homo sapiens*) | HEK293T | ATCC | CRL-3216 | |
| Cell line (*Homo sapiens*) | CAP-T | *Fischer et al., 2012*, Biotechnology and bioengineering 109, 2250–2261, 2012 | | |
| Cell line (*Rattus norvegicus, Mus musculus*) | 2.4G2 | ATCC | HB-197 | Hybridoma cell line, *Rattus norvegicus* (B-cell), *Mus muculus* (myeloma) |
| Other | BALB/c mouse | Charles River | Mouse; *Mus musculus*; Strain code: 028 | 10–12 weeks old; $n = 6$/group (4× female, 2× male) |
| Peptide, recombinant protein | SARS-CoV-2 spike peptide-loaded dimer | JPT Peptide Technologies GmbH | | CoVK$^{d-268}$ GYLQPRTFL |
| Recombinant DNA reagent | pBSK-CMV-Spike | This paper | Based on GenBank sequence YP_009724390.1 | CMV promoter-driven SARS-CoV-2 spike protein with some modifications |
| Antibody | Rat anti- murine-INFγ-PE, monoclonal | Invitrogen | 12-7311-82 | IF 1:200 |
| Antibody | Rat anti-murine-CD8-APC, monoclonal | eBioScience | 17-0081-83 | IF 1:200 |
| Antibody | Rabbit anti-mouse IgG, HRP-labeled, polyclonal | Sigma | A9044 | ELISA: 1:10,000 |
| Antibody | Rat/mouse anti-Fc gamma Receptor (FcRII, CD32), monoclonal | Derived from hybridoma 2.4G2 cell line (ATCC HB-197) | | Four-day-old cell culture medium, undiluted |
| Chemical compound, drug | Brefeldin A | eBioScience | 00-4506-51 | |
| Chemical compound, drug | Ni-NTA agarose beads | Qiagen | 30,210 | |
| Sequence-based reagent | Adenoviral *E4* forward | GeneArt Sequences | PCR primer | TAGACGATCCCTACTGTACG |
| Sequence-based reagent | Adenoviral *E4* reverse | GeneArt Sequences | PCR primer | GGAAATATGACTACGTCCGG |

*Continued on next page*

*Continued*

| Reagent type (species) or resource | Designation | Source or reference | Identifiers | Additional information |
|---|---|---|---|---|
| Sequence-based reagent | Adenoviral *human actin* forward | GeneArt Sequences | PCR primer | GCTCCTCCTGAGCGCAAG |
| Sequence-based reagent | Adenoviral *human actin* reverse | GeneArt Sequences | PCR primer | CATCTGCTGGAAGGTGGACA |
| Sequence-based reagent | Adenoviral *human ribosomal protein L4* forward | GeneArt Sequences | PCR primer | ACGATACGCCATCTGTTCTGCC |
| Sequence-based reagent | Adenoviral *human ribosomal protein L4* reverse | GeneArt Sequences | PCR primer | GGAGCAAAACAGCTTCCTTGGTC |
| Chemical compound, drug | SYBR Green | Kapa Biosystems | KK4502 | |
| Commercial assay or kit | ADP FI Assay | **Bos et al., 2020**, Bellbrooks Labs | 3013A | |
| Commercial assay or kit | GenElute Mammalian Genomic DNA Miniprep Kit | Sigma | G1N350 | |
| Commercial assay or kit | VenorGeM Classic | Minerva Biolabs | 11-1025 | |
| Software, algorithm | RStudio | RStudio | Version 4.0.0 | Statistics |
| Software, algorithm | MaxQuant (with Andromeda search engine; UniProt human reference proteome) | **Cox and Mann, 2008**; **Cox et al., 2011**, MaxQuant, Andromeda,UniProt | Version 1.6.3.4 | Protein identification Mass Spectrometry |

## Adenoviral vectors and vaccines

All vaccine lots analyzed in this study were obtained in original vials from the Pharmacy of the University Hospital Ulm between March 2021 and September 2021, as they became available in the course of the ongoing vaccination campaign.

According to the manufacturer, ChAdOx1 (lots ABV4678, ABV5811, ABV7764, and ABV9317) has a physical titer of $1 \times 10^8$ VP/µl and is dissolved in 10 mM histidine, 7.5% sucrose (wt/vol), 35 mM NaCl, 1 mM $MgCl_2$, 0.1% polysorbate 80 (wt/vol), 0.1 mM Ethylenediamine tetraacetic acid (EDTA), and 0.5% EtOH (wt/vol) (**European-Medicines-Agency, 2021a**).

According to the manufacturer, Ad26.COV2.S (lots XD955, 21C10-01, XE395) has a physical titer of $1 \times 10^8$ VP/µl and is dissolved in citric acid monohydrate (0.14 mg), trisodium citrate dehydrate (2.02 mg), ethanol (2.04 mg), 2-hydroxypropyl-β-cyclodextrin (25.50 mg), polysorbate-80 (0.16 mg), and sodium chloride (2.19 mg) (**Janssen-Pharmaceutical-Companies, 2021**).

HAdV-C5-EGFP vector particles used in this study are *E1*-deleted replication-incompetent vector particles of human adenovirus species C type 5 (based on GenBank AY339865.1, sequence from nt 1 to 440 and from nt 3523 to 35,935). HAdV-C5-EGFP carries a CMV promoter-controlled EGFP expression cassette, which was subcloned from a pEGFPN1 plasmid (6085-1; Clontech).

UUlm ChAdOx1 vector particles were derived from AstraZeneca ChAdOx1 nCov-19 vaccine lot ABV4678. HAdV5-C5-EGFP and UUlm ChAdOx1 particles were produced in *E1*-complementing N52. E6 cells (**Schiedner et al., 2000**) or HEK293T cells, respectively, as previously described (**Nilson et al., 2021**). In brief, $2 \times 10^8$ cells were transduced with $6 \times 10^{10}$ total vector particles from stock solution. Forty-eight hours post-transduction cells were harvested, resuspended in 3 ml buffer (50 mM 4-[2-hydroxyethyl]-1-piperazineethanesulfonic acid [HEPES], 150 mM NaCl, pH 7.4) and lysed by three consecutive freeze/thaw cycles. Cell debris was removed by centrifugation at 2000 × *g* for 10 min, and vector particle-containing supernatants were layered on a CsCl step gradient (density bottom: 1.41 g/ml; density top: 1.27 g/ml, 50 mM HEPES, 150 mM NaCl, pH 7.4) and centrifuged at 176,000 × *g* for 2 hr at 4°C. Vector particles were further purified by a consecutive continuous CsCl gradient (density: 1.34 g/ml, 50 mM HEPES, 150 mM NaCl, pH 7.4) and centrifuged at 176,000 × *g* for 20 hr at

4°C. Subsequently, vector solutions were desalted by size exclusion chromatography (PD10 columns, 17-0851-01; GE Healthcare). Physical vector titers were determined by optical density measurement at 260 nm as described earlier (*Mittereder et al., 1996*).

HAdV-C5-EGFP and UUlm ChAdOx1 vectors, produced in-house, had physical titers of $2.9 \times 10^9$ and $1.65 \times 10^8$ VP/µl, respectively, and were dissolved in 50 mM HEPES, 150 mM NaCl, 10% glycerol, pH 7.4.

## Cell lines

2.4G2 cells (ATCC, HB-197) were split 3×/week and maintained in RPMI medium (Gibco, 21875-034). HEK293T cells (ATCC, CRL-3216) were split 2×/week and maintained in Dulbeccos's Modified Eagle Medium (DMEM) (Gibco, 41966-029). N52.E6 cells (*Schiedner et al., 2000*) were split 2×/week and maintained in alphaMEM medium (Gibco, 22561-021). All media were supplemented with 10% Fetal Calf Serum (FCS) and 1% penicillin/streptomycin/glutamine and cells were cultivated in cell culture dishes (Nunc, 168381) at 37°C in 5% $CO_2$ and a relative humidity of 95%.

CAP-T cells (*Fischer et al., 2012*), kindly provided by CEVEC Pharmaceuticals GmbH, Cologne, for research purposes, were cultivated in Protein Expression Medium (PEM) (Gibco, 12661-013) with 4 mM Glutamax in polycarbonate plain shake flasks at 185 rpm and 37°C in 5% $CO_2$ with a relative humidity of 95%. Cells were split 2×/week to a density of $1 \times 10^6$ cells/ml.

All cell lines used in this study were tested negative for mycoplasma contamination (VenorGeM Classic Minerva Biolabs, 11-1025).

## Protein staining after SDS–PAGE

$3 \times 10^9$ vector particles dissolved in 30 µl were mixed with SDS-loading buffer (30 mM Tris, 1% SDS, 5% glycerol, bromophenol blue, pH 7.5) containing 0.2 M β-mercaptoethanol and heated for 5 min at 96°C. Reduced and denatured proteins were separated by SDS–PAGE and subsequently stained in gel by Coomassie or silver staining. For Johnson & Johnson Ad26.COV2.S vector storage buffers were exchanged before SDS–PAGE using 30 kDa cutoff amicons (Millipore, MPUFC5030BK) according to the manufacturer's instructions after heating samples for 5 min at 96°C.

Silver staining: proteins were fixed (50% MeOH, 12% AcOH, 0.05% HCHO) for 30 min and washed for 15 min with 50% EtOH. Subsequently, proteins were equilibrated for 1 min (0.8 mM $Na_2S_2O_3$), washed with $dH_2O$, impregnated for 20 min (11.78 mM $AgNO_3$, 0.05% HCHO) and washed again with $H_2O$. Protein bands were visualized by the deoxidation of adsorbed silver ions to silver (0.57 M $Na_2CO_3$, 0.05% HCHO, 15.8 µM $Na_2S_2O_3$). Signal development was stopped (50% MeOH, 12% AcOH) once protein bands were visible.

Coomassie staining: protein gels were stained by Coomassie (3.5 mg/ml Coomassie Brillant Blue R-250, 30% EtOH, 10% AcOH) for 12 hr. Subsequently, the gel was destained (30% EtOH, 10% AcOH, exchanged every 20 min) for 2 hr.

## Determination of protein amounts

Protein content in ChAdOx1 nCov-19 was determined by classical methods: NanoDrop-based absorbance measurement at 280 nm, Bradford Assay, and protein staining intensities.

### Absorption at 280 nm

ChAdOx1 lots were heated for 5 min at 70°C to inactive and disassemble vector particles. ChAdOx1 buffer according to the manufacture was used as a blank reference. Samples were analyzed using NanoDrop OneC (Thermo Fisher) according to the manufacturer's protocol. Absorption at 280 nm was used employing the 'A280 mg/ml' routine embedded in the instrument (Ver. 1.3, DB version 1).

### Bradford assay

100 µl 1:5 diluted ProteinAssay Dye (BioRad Laboratories, 50000006) was mixed with 4 µl of ChAdOx1 nCov-19 lots. Bovine serum albumin (BSA) dissolved in ChAdOx1 buffer with concentrations ranging from 2 to 0 mg/ml was used as reference. Samples were thoroughly mixed and measured in triplicates at 560 nm using an ELISA Reader (Mulitscan Ex, Thermo Fisher Scientific).

Optical determination of protein signal intensities in gel: upon silver staining of SDS–PAGE separated proteins, signal intensities of ChAdOx1 nCov-19 lots were determined using ImageJ 1.53a

software and quantities calculated relative to HAdV-C5-EGFP, corresponding to 12.8 µg protein for 5 × $10^{10}$ vector particles/dose.

## Sample preparation for proteomic analysis

For sample clean-up, 0.5 ml vector solution was precipitated employing methanol/chloroform extraction. Two ml methanol was added and mixed, 500 µl chloroform added and the mixture thoroughly vortexed. After addition of 1.5 ml of water and mixing, suspension was centrifuged for 1 min at 14,000 × $g$. The top layer was removed and another 2 ml of methanol added. Following centrifugation for 5 min at 14,000 × $g$, methanol was removed and the pellet collected for further analysis.

## Proteomic analysis of the ChAdOx1 and Ad26.COV2.S vaccine

For in-solution digests, 6 µg of protein was reduced with 5 mM dithiothreitol (DTT) (AppliChem) for 20 min at RT and subsequently alkylated with iodoacetamide (Sigma-Aldrich) for 20 min at 37°C. Trypsin (Thermo Scientific) was added in a 1:50 enzyme–protein ratio and digested overnight at 37°C. Samples separated via SDS–PAGE were stained by Coomassie. Each gel lane was cut into 20 pieces and prepared for LC/MS analysis as described (*Hecht et al., 2019*). LC/MS and bioinformatical analysis were carried out as described above, with the exception of shortening the LC gradient to 65 min in total. Employing an LTQ Orbitrap Elite system (Thermo Fisher Scientific) online coupled to an U3000 RSLCnano (Thermo Fisher Scientific), samples were analyzed as described (*Mohr et al., 2015*), with modifications (*Hecht et al., 2019*) and picking the 20 most intense ions from the survey scan for CID fragmentation. Singly charged ions were rejected and *m/z* of fragmented ions were excluded from fragmentation for 60 s. MS2 spectra were acquired employing the LIT at rapid scan speeds.

## MS data analysis and statistics

Database search was performed using MaxQuant Ver. 1.6.3.4 (*Cox and Mann, 2008*). Employing the build-in Andromeda search engine (*Cox et al., 2011*), MS/MS spectra were correlated with the UniProt human reference proteome set (https://www.uniprot.org/) and a database containing the expected virus protein sequences for peptide identification. Carbamidomethylated cysteine was considered as a fixed modification along with oxidation (M), and acetylated protein N-termini as variable modifications. False discovery rates were set on both, peptide and protein level, to 0.01.

## Detection of adenoviral and human genome DNA by qPCR

DNA of 1 × $10^{11}$ vector particles of HAdV-C5-EGFP and AstraZeneca ChAdOx1 nCoV-19 was isolated using GenElute Mammalian Genomic DNA Miniprep Kit (Sigma, G1N350) according to the manufacturer's instructions. As a control, DNA of 2 × $10^{6}$ HEK293T cells (ATCC CRL-3216) was isolated. DNA was eluted in 10 mM Tris, pH 8.5. Concentration was determined by optical density measurement at 260 nm. DNA samples were analyzed for their viral and human genome DNA content by quantitative real time PCR. Primers used amplified parts of the adenoviral E4 region, which is present in both adenoviral strains analyzed (forw.: 5′ TAGACGATCCCTACTGTACG 3′; rev.: 5′ GGAAATATGACTACGTCCGG 3′), the human actin gene (forw.: 5′ GCTCCTCCTGAGCGCAAG 3′; rev.: 5′ CATCTGCTGGAAGGTGGACA 3′), and the human ribosomal protein L4 gene (forw.: 5′ ACGATACGCCATCTGTTCTGCC 3′; rev.: 5′ GGAGCAAAACAGCTTCCTTGGTC 3′). 20 ng DNA was added to 10 µl SYBR Green (KK4502; Kapa Biosystems), and 0.4 µl 10 pmol/µl of each forward and reverse primer in a total volume of 20 µl. Thermocycles: 1 cycle: 10 min 95°C; 40 cycles: 30 s 95°C, 30 s 60°C, 8 s 72°C; 1 cycle: 10 min 72°C.

## Determination of ATPase activity

ATPase activity in AstraZeneca ChAdOx1 nCoV-19 vaccine was measured using the $ADP^{2}$ () FI Assay (Bellbrooks Labs, 3013 A), which gives increasing fluorescence intensity signals with increasing amounts of ADP. 20 µl AstraZeneca ChAdOx1 nCoV-19 lot ABV9317 were mixed with final concentrations of 4 mM $MgCl_2$ and 10 µM ATP in a final volume of 25 µl in black 96-well plates (Corning, 3881) in triplicates. Samples were incubated for 30 min at 37°C before 25 µl Stop&Detect solution (10 µg/ml ADP-antibody-IRDye-QC1, 8 nM ADP-Alexa Fluor594 Tracer, 1× stopping buffer) was added. Samples were incubated for 1 hr at RT in the dark. Samples were mixed for 20 s at 480 rpm and fluorescence

signal measured in 96-well plate reader. Settings: central Ø2 mm area read; fixed gain: 2400; fixed focal height: 4 mm.

Negative controls: 50 mM HEPES instead of ABV9317; without addition of 10 µM ATP. Positive control: addition of 10 µM ADP instead of 10 µM ATP.

## Animal experiments

Ten- to twelve-week-old BALB/c mice (Charles River) were maintained in pathogen-free, individually ventilated cages, and fed with sterilized diet for laboratory rodents. All experiments were in accordance with policies and procedures of institutional guidelines and approved by the Animal Care Commission of the Government Baden-Württemberg. Animals were briefly anesthetized with sevoflurane and injected intramuscularly with phosphate-buffered saline (PBS) or $1 \times 10^9$ vector particles of AstraZeneca ChAdOx1 nCoV-19 vaccine ABV9317 or UUlm ChAdOx1 dissolved in sterile PBS in final volumes of 100 µl (50 µl/*M. gastrocnemii*). To exclude effects induced by AstraZeneca ChAdOx1 nCoV-19 vaccine buffer components, UUlm ChAdOx1 samples were supplemented with equal amounts of the buffer. Fourteen days after injection, animals were sacrificed by overdosed sevoflurane anesthesia and subsequent mechanical rupture of the diaphragm, blood samples were collected and anticoagulated with 1 U/ml heparin. Plasma was prepared by centrifugation of blood samples for 45 min at 800 × *g* and stored at −20°C. Additionally, spleens were isolated and directly used for the determination of Spike-specific T-cell responses. *n* = 6/group (4× female, 2× male).

## Analysis of T-cell responses

Murine spleens were dissociated using strainers in PBS with 1% BSA and centrifuged for 5 min at 300 × *g*. To lyse erythrocytes, cell pellets were resuspended in Tris-buffered ammonium chloride lysis buffer and incubated for 4 min. Cells were centrifuged for 5 min at 300 × *g*, washed with 10 ml PBS/1% BSA and resuspended in 3 ml Ultraculture medium (lonza/Biozym, 88100). 50 µl cell suspension was mixed with 25 µl of spike peptide-loaded dimer CoVK[d-268] GYLQPRTFL (JPT Peptide Technologies GmbH) in a final concentration of 10 µg/ml and 25 µl Brefeldin A (eBioScience, 00-4506-51) and incubated for 5 hr at 37°C, followed by incubation overnight at 4°C. The next day, cells were centrifuged for 5 min at 300 × *g* and pellets resuspended in buffer A (PBS, 0.5% BSA, 0.1% NaN₃). 100 µl murine FcR block solution (cell culture supernatant of 2.4G2 cells) was added and incubated for 15 min at 4°C. Subsequently 50 µl of anti-murineCD8-APC antibody (eBioScience, 17-0081-83, diluted 1:200 in FcR blocking solution) and cells were incubated for 30 min at 4°C. Cells were washed twice with 150 µl buffer A before 150 µl 2% paraformaldehyde (PFA) dissolved in buffer A was added and cells incubated for 15 min at 4°C. Cells were washed once with buffer A, resuspended in buffer B (PBS, 0.5% BSA, 0.05% NaN₃, 0.35% saponine) and incubated for 15 min at 4°C to perforate the cells. Cells were centrifuged, pellets were resuspended with 50 µl anti-murineINFγ-PE antibody (Invitrogen, 12-7311-82, diluted 1:200 in buffer B) and incubated for 30 min at 4°C. Cells were washed once with buffer A, resuspended in 100 µl buffer A and analyzed by flow cytometry for APC/PE double-positive cells.

## Analysis of antibody responses

Total spike-specific antibody titers were determined by ELISA. In brief: 100 µl of 0.5 µg/ml purified spike protein dissolved in coating buffer (0.04 M Na₂CO₃, 0.06 M Na₂HCO₃, pH 9.6) was coated on Maxisorp ELISA plates overnight at 4°C. Upon blocking of wells for 1 hr at RT (Thermo Fisher, 37535), 100 µl murine plasma samples diluted in PBS (starting dilution: 1:20; 1:5 serial dilutions) was added and incubated for 2 hr at RT. Subsequent 100 µl HRP-labeled rabbit anti-mouse antibody was added (Sigma A9044-2ML; dilution 1:10.000) and incubated for 1 hr at RT. Chemiluminescence signals were induced by adding 100 µl TMB substrate (Thermo Fisher, N301) and stopped by addition of 100 µl 2 M H₂SO₄. Signals were measured at 450 nm.

## Purification of SARS-CoV-2 spike protein

A CMV-promoter controlled SARS-CoV-2 Spike expression plasmid based on GenBank sequence YP_009724390.1 (modifications: deleted transmembrane domain, µ phosphatase signal peptide, C-terminal 8xHis-Tag, artificial trimerization domain, two proline insertions, deleted furin cleavage site, SV40 poly(A), codon-optimized for *Homo sapiens*) was used for transfection of CAP-T cells (*Fischer et al., 2012*). 96 hr p.t. cells were separated and 30 ml of supernatant were concentrated

to 250 µl using 100 kDa cutoff amicons (Millipore, UFC9100) by centrifugation for 5 hr at 4600 rpm. Concentrates were resuspended in buffer A (50 mM NaH$_2$PO$_4$, 300 mM NaCl, 20 mM imidazole, 0.05% Tween20, 1× protease inhibitor, pH 8.0) and 100 µl Ni-NTA agarose beads (Qiagen, 30210) in a total volume of 2 ml and incubated overnight at 4°C, rotating. Using Poly-Prep Chromatography Columns (BioRad Laboratories, 7311550) beads were thoroughly washed and spike protein eluted with 500 µl buffer B (50 mM NaH$_2$PO$_4$, 300 mM NaCl, 250 mM imidazole, 1× protease inhibitor, pH 8.0). The purity of isolated spike protein was confirmed by silver staining of proteins separated by SDS–PAGE.

## Statistics

Statistical analysis of samples was performed using unpaired two-sample Student's *t*-test with RStudio-software Version 4.0.0. p values ≤0.05 were considered statistically significant.

## Acknowledgements

We thank Prof. Reinhold Schirmbeck, Dr. Katja Stifter, Prof. Holger Barth and the members of the Department of Gene Therapy for discussion and Dr. Ludwig Maier and Prof. Guido Adler for support. We thank Prof. Thomas Stamminger, Prof. Frank Kirchhoff, Prof. Jan Münch, and Prof. Dierk Niessing for internal review. We thank CEVEC Pharmaceuticals for providing CAP-T cells for research studies and Dr. Keller-Stanislawski (PEI) and Prof. Eberhard Hildt (PEI) for advice. The work was supported by the German Research Foundation (SFB1074) and by the German Federal Ministry of Education and Research (BMBF) and the Federal States of Germany Grant 'Innovative Hochschule' (FKZ 3IHS024D).

## Additional information

### Funding

| Funder | Grant reference number | Author |
| --- | --- | --- |
| Bundesministerium für Bildung und Forschung | Grant "Innovative Hochschule" FKZ3IHS024D | Lea Krutzke<br>Reinhild Rösler<br>Ellen Allmendinger<br>Tatjana Engler<br>Sebastian Wiese<br>Stefan Kochanek |
| German Research Foundation | SFB1074 | Lea Krutzke<br>Reinhild Rösler<br>Ellen Allmendinger<br>Tatjana Engler<br>Sebastian Wiese<br>Stefan Kochanek |

The funders had no role in study design, data collection, and interpretation, or the decision to submit the work for publication.

### Author contributions

Lea Krutzke, Conceptualization, Data curation, Formal analysis, Investigation, Methodology, Resources, Supervision, Visualization, Writing - original draft, Writing - review and editing; Reinhild Rösler, Data curation, Formal analysis, Investigation, Methodology, Writing - original draft, Writing - review and editing; Ellen Allmendinger, Tatjana Engler, Investigation, Methodology, Writing - review and editing; Sebastian Wiese, Data curation, Formal analysis, Funding acquisition, Supervision, Validation, Writing - original draft, Writing - review and editing; Stefan Kochanek, Conceptualization, Formal analysis, Funding acquisition, Investigation, Project administration, Resources, Supervision, Writing - original draft, Writing - review and editing

### Author ORCIDs

Lea Krutzke ⓘ http://orcid.org/0000-0002-4092-4131
Stefan Kochanek ⓘ http://orcid.org/0000-0001-7494-1602

### Ethics

Animal experiments were approved by the Animal Care Commission of the Government Baden-Wü;rttemberg. Reference number: TVA #1508.

### Decision letter and Author response

Decision letter https://doi.org/10.7554/eLife.78513.sa1

Author response https://doi.org/10.7554/eLife.78513.sa2

---

## Additional files

### Supplementary files

• MDAR checklist

### Data availability

All data supporting the findings of this study are available within this paper. An overview of protein identifications and quantifications based on LC/MS analysis is shown in the source data (Figure 2—source data 1, Figure 3—source data 1, and Figure 4—source data 1). LC/MS-raw data and search results have been deposited at the Mass Spectrometry Interactive Virtual Environment (MassIVE; https://massive.ucsd.edu/ProteoSAFe/static/massive.jsp) data lake and are publicly available under ID MSV000089634.

The following dataset was generated:

| Author(s) | Year | Dataset title | Dataset URL | Database and Identifier |
|---|---|---|---|---|
| Stefan K | 2022 | Process- and product-related impurities in the ChAdOx1 nCov-19 vaccine | https://doi.org/doi:10.25345/C50G3H28F | MassIVE, 10.25345/C50G3H28F |

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
