## [Editor Report]

This research shows that a commonly used commercial vaccine for COVID-19 harbors contaminating proteins derived from the human cell line in which it is produced. The health significance of these contaminants (if any) remains unknown. This paper is important because lot purity and processing of vaccines is rarely scrutinized in the scientific realm, and instead is typically analyzed only by the companies themselves.

---

## [Decision Letter]

**Decision letter after peer review:**

Thank you for submitting your article "Process- and product-related impurities in the ChAdOx1 nCov-19 vaccine" for consideration by *eLife*. Your article has been reviewed by 4 peer reviewers, and the evaluation has been overseen by myself as a Reviewing Editor and Diane Harper as the Senior Editor. The following individuals involved in the review of your submission have agreed to reveal their identity: Thomas Klimkait (Reviewer #1); Roland Wagner (Reviewer #3).

Essential revisions:

We will not require further experiments, but you can see from the reviews that many questions arose throughout. Given the interest that the paper will garner, we ask you to address and clarify as many of these points as possible. In particular, please add a description of how these lots were chosen. Please also add as much detail as possible regarding what is known about the differences in the purification process of ChAdOx1 versus Ad26.COV2.S.

*Reviewer #1 (Recommendations for the authors):*

The manuscript provides strong evidence for a surprisingly high heterogeneity in protein content when comparing 3 preparations of a commercial vaccine preparation. The authors appropriately use widely used and standardised methods for the elucidation of the protein composition of several Aden-based vaccine preparations. After the somewhat unexpected finding of high protein contamination levels in ChAdOx1, they apply different ways to verify exceedingly high levels of contamination with host cell protein. As perplexing as a 70% contamination may sound, the authors plausibly discuss possible reasons for the massive under quantification.

It is highly astonishing that no response by authorities (EMA) or manufacturer (Astra) is found in the manuscript. To the reader, this indicates that those bodies either are not aware of the study data or that they did not respond. It might therefore be of high general interest to add a short statement to the discussion.

Overall, the study has been carefully designed and conducted, and the latter part including platelet-related ATPase experiments adds important detail to the discussion about potential product-induced side effects after vaccination.

The manuscript is clearly structured and the analyses are straightforward. The detailed discussion of a possible side-effect potential caused by contaminating cell proteins adds important information.

Also the information about why certain proteins, i.e. HSP could have been missed although present at extreme levels, provides very useful information for the non-expert audience.

*Reviewer #2 (Recommendations for the authors):*

This is a simple study looking at the impurities of two widely used Covid-19 vaccines, ChAdOx1 and Ad26.COV2.S. The result that all four lots of ChAdOx1 analyzed contained impurities above the maximum allowable level was surprising and important.

1) I would like to see the authors include as much detail as possible regarding what is known about the differences in the purification process of ChAdOx1 versus Ad26.COV2.S. It would be tremendously helpful for the field to understand what part of the ChAdOx1 purification process is responsible for the high levels of impurities.

2) Why is the Penton base band for ChAdOx1 not visible in Figure 1 and Figure 3A?

3) I would have liked to have seen all of the analyses performed in Figure 3 to have been conducted on HAdV-C5-EGFP and on UUlm ChAdOx1.

To improve the usefulness of the manuscript, the authors should discuss in more detail the possible reasons for the high levels of impurities so that this can be avoided in the future.

*Reviewer #3 (Recommendations for the authors):*

The elimination of endogenously produced host cell proteins is one of the most challenging operations in guaranteeing the compliance of critical quality attributes in the production of biological drug substances. Realization of process robustness sufficient enough to consistently keep it at the highest standard is a time-consuming part of the process development. Hence, it is of utmost importance to have a very critical view of drugs that had been developed and produced under conditions of a propagating disease wave and an increasing political and social pressure.

Compromising the purity of a drug with host cell proteins is most probable when the product is produced by the biological system of a cell. This is especially the case for the vaccines against SARS-CoV-2 infection based on adenoviruses as vectors because the virus has to be produced by means of animal cells. Therefore, the authors investigated the two approved vaccines produced by AstraZeneca and Johnson and Johnson. The manuscript is clearly written and the methods used for analyzing the impurities are industrial standard methods used in the process development of therapeutic bioproducts. The authors have long-term expertise in adenovirus production and development. The conclusive results are highly reasonable.

Very important investigation and very important results that have to be published as fast as possible. The manuscript is clearly written and the methods used for analyzing the impurities are reliable standard methods used for identifying impurities in drug-producing bioprocesses. The conclusive results are highly reasonable and such that I recommend the manuscript for publication without reservations.

*Reviewer #4 (Recommendations for the authors):*

1. I did not find a description of how these lots were chosen. Ie: I am guessing there are many lots out there and these three a random choice? (why only 3?). There are then statements like (line 319) 'ChAdOx1 but not Ad26.COV2.S contained high amounts of human and free viral proteins'. When I understand that one Ad26 contained less than 3 AZ. Some statistical comparison of a larger number of randomly sampled lots would seem appropriate?

2. The immunological analysis seems pretty superficial and also subject to type 2 error. Surely the statement should be 'responses to AZ were 3%, and to control 2%, although this was not significant (stating p-value and test, not currently stated)' (rather than ignoring difference). Same for spike binding antibodies. Similarly, the discussion pointed out a trend of high antibodies + T cells that was not significant. Overall this is a very superficial look at immune responses.

3. There is a lot of excessive speculation in the discussion. Eg:

a) Line 281: "we assume….might enhance early clinical vaccine reactions".

b) Line 289 'might be more than inert bystanders…'

c) 296: 'transfer of peptides'.

d) 307 onwards: speculation on VIIT. Shouldn't this just say no PF4 was found…

e) Line 33 onwards – very little immune data, and the data they have suggests higher immunogenicity (but not significant).

f) Lines 341: They speculate on antibodies to the impurities. It is pretty easy to check – so not sure why speculation?

g) The speculation about the possible effects of contaminants cites reference 29 extensively (which looks at the impact of contaminants). My reading of this is that (i) most of the documented cases come from bacterial contaminants (that act as adjuvants). (ii) the major effect is immune responses that reduce the circulation time/effects of a drug. (iii) there seem few documented cases where this occurs, and none where there are autoimmune consequences?

h) (line 349) bovine neonatal pancytopenia (BNP): no citations provided, but also not clear it is relevant (ie: it needs a statement that no MHC molecules found in the vaccines, no evidence for alloreactivity, etc – so this does not seem relevant to the case at hand)

I feel it is written in a way to enhance speculation on the role of contaminants, whereas in fact there seems no plausible link with any side effect (and the most obvious mechanisms – such as contamination with PF4) can be eliminated by the study…

I think the manuscript would be greatly enhanced if the authors could:

1) to show clear evidence of whether there is a difference in contamination between this and other vaccines (esp JnJ). Eg: sufficient (randomly chosen) batches to show statistical significance (or a clear statement that based on 1 batch versus 3 this is purely anecdotal).

2) Before speculating on auto-antibodies etc, this should be directly examined (eg: look at antibodies from AZ / JnJ vaccinated and control individuals to cell lysate (western blot? Immunoprecipitation?). These are very easy. Also, make recombinant versions of the top 5 contaminants and see if antibodies to these are detectable in AZ vaccinated vs. control vs. JnJ vaccinated).

3) It should clearly state that although there is a theoretical risk potential contaminants could have an effect, no relevant examples are known in the literature (not my area – but going off reference 29) and no relevant proteins were found. Eg: no PF4 (relevant to HIIT), no MHC (relevant to bovine neonatal pancytopenia).

---

## [Author Response]

Essential revisions:We will not require further experiments, but you can see from the reviews that many questions arose throughout. Given the interest that the paper will garner, we ask you to address and clarify as many of these points as possible. In particular, please add a description of how these lots were chosen. Please also add as much detail as possible regarding what is known about the differences in the purification process of ChAdOx1 versus Ad26.COV2.S.

We have addressed/clarified the reviewers’ questions, as detailed in the specific comments to the reviewers below.

Lines 377-380: description, how the lots were obtained

Lines 78-81: a comparison between the purification processes of ChAdOx1 and Ad26.COV2.S is not possible, since only in case of the AZ vaccine, detailed information on the manufacturing is available (new reference Joe et al., 2022). On the manufacturing of the JJ vaccine no details of upstream and downstream processes are publicly available. However, reference Joe et la., 2022 provides important information on the process in case of ChAdOx1.

Reviewer #1 (Recommendations for the authors):The manuscript provides strong evidence for a surprisingly high heterogeneity in protein content when comparing 3 preparations of a commercial vaccine preparation. The authors appropriately use widely used and standardised methods for the elucidation of the protein composition of several Aden-based vaccine preparations. After the somewhat unexpected finding of high protein contamination levels in ChAdOx1, they apply different ways to verify exceedingly high levels of contamination with host cell protein. As perplexing as a 70% contamination may sound, the authors plausibly discuss possible reasons for the massive under quantification.It is highly astonishing that no response by authorities (EMA) or manufacturer (Astra) is found in the manuscript. To the reader, this indicates that those bodies either are not aware of the study data or that they did not respond. It might therefore be of high general interest to add a short statement to the discussion.

The regulatory bodies (EMA, PEI) and the manufacturer (AZ) were immediately informed on the results of our analysis of the AZ vaccine, as soon as the data were obtained.

Since this is not scientific information, we prefer not to introduce this information into the manuscript.

Reviewer #2 (Recommendations for the authors):This is a simple study looking at the impurities of two widely used Covid-19 vaccines, ChAdOx1 and Ad26.COV2.S. The result that all four lots of ChAdOx1 analyzed contained impurities above the maximum allowable level was surprising and important.1) I would like to see the authors include as much detail as possible regarding what is known about the differences in the purification process of ChAdOx1 versus Ad26.COV2.S. It would be tremendously helpful for the field to understand what part of the ChAdOx1 purification process is responsible for the high levels of impurities.

Lines 78-81: a comparison between the purification processes of ChAdOx1 and Ad26.COV2.S is not possible, since only in case of the ChAdOx1 vaccine detailed information on the manufacturing is available (new reference Joe et al., 2022 is now cited). We note that compared to a previous publication (reference Fedosyuk et al., 2019) a TFF step was omitted prior to the anion-exchange chromatography step to facilitate scale-up. However, without more detailed information it would be too speculative to link this modification to the high HCP content observed in the vaccine. Therefore, we have just indicated that modifications of the process have been introduced. The interested reader can look up reference 5 for more details.

On manufacturing of the JJ vaccine no details of upstream and downstream processes are publicly available, only what can be found at the EMA (Reference European-Medicines-Agency, 2021b).

2) Why is the Penton base band for ChAdOx1 not visible in Figure 1 and Figure 3A?

Very good observation. Same is true for the Ad26.COV2.S

Briefly: Penton Base of HAdV-C5 is larger than of ChAdOx1 and Ad26.COV2.S, and the disorder of the RGD-containing loop in HAdV-C5 is more disorders.

Lines 784-792: Detailed explanation has been introduced in the legend of Figure 3.

3) I would have liked to have seen all of the analyses performed in Figure 3 to have been conducted on HAdV-C5-EGFP and on UUlm ChAdOx1.

All analyses are now shown also for the additional 2 lots of Ad26.COV2.S (Figure 3 —figure supplement 2), for HAdV-C5-EGFP (Figure 3 —figure supplement 1) and for UUlm ChAdOx1 (Figure 4 —figure supplement 1). The corresponding source data are also included.

To improve the usefulness of the manuscript, the authors should discuss in more detail the possible reasons for the high levels of impurities so that this can be avoided in the future.

Lines 78-81: a comparison between the purification processes of ChAdOx1 and Ad26.COV2.S is not possible, since only in case of the ChAdOx1vaccine detailed information on the manufacturing is available (new reference Joe et al., 2022 is now cited). We note that compared to a previous publication (reference Fedosyuk et al., 2019) a TFF step was omitted prior to the anion-exchange chromatography step to facilitate scale-up. However, without more detailed information it would be too speculative to link this modification to the high HCP content observed in the vaccine. Therefore, we have just indicated that modifications of the process have been introduced. The interested reader can look up reference 5 for more details.

On manufacturing of the JJ vaccine no details of upstream and downstream processes are publicly available, only what can be found at the EMA (Reference European-Medicines-Agency, 2021b).

Reviewer #4 (Recommendations for the authors):1. I did not find a description of how these lots were chosen. Ie: I am guessing there are many lots out there and these three a random choice? (why only 3?). There are then statements like (line 319) 'ChAdOx1 but not Ad26.COV2.S contained high amounts of human and free viral proteins'. When I understand that one Ad26 contained less than 3 AZ. Some statistical comparison of a larger number of randomly sampled lots would seem appropriate?

Choice of lots: All vaccine lots analyzed in this study were obtained in original vials from the Pharmacy of the University Hospital Ulm between March 2021 and September 2021, as they became available in the course of the ongoing vaccination campaign

There has been misunderstanding: a total of 4 AZ (not 3) vaccines and 3 JJ (not 1) vaccines were analysed and all data, including source data, are now presented in the manuscript. This has been explained now in a clearer way in the manuscript.

2. The immunological analysis seems pretty superficial and also subject to type 2 error. Surely the statement should be 'responses to AZ were 3%, and to control 2%, although this was not significant (stating p-value and test, not currently stated)' (rather than ignoring difference). Same for spike binding antibodies. Similarly, the discussion pointed out a trend of high antibodies + T cells that was not significant. Overall this is a very superficial look at immune responses.

As we mention in the manuscript, the study was meant as a pilot study.

Lines 214 – 222. We have now modified the wording according to the reviewer’s suggestion to more accurately describe our findings.

3. There is a lot of excessive speculation in the discussion. Eg:a) line 281: "we assume….might enhance early clinical vaccine reactions".b) Line 289 'might be more than inert bystanders…'c) 296: 'transfer of peptides'.d) 307 onwards: speculation on VIIT. Shouldn't this just say no PF4 was found…e) Line 33 onwards – very little immune data, and the data they have suggests higher immunogenicity (but not significant).f) Lines 341: They speculate on antibodies to the impurities. It is pretty easy to check – so not sure why speculation?g) The speculation about the possible effects of contaminants cites reference 29 extensively (which looks at the impact of contaminants). My reading of this is that (i) most of the documented cases come from bacterial contaminants (that act as adjuvants). (ii) the major effect is immune responses that reduce the circulation time/effects of a drug. (iii) there seem few documented cases where this occurs, and none where there are autoimmune consequences?h) (line 349) bovine neonatal pancytopenia (BNP): no citations provided, but also not clear it is relevant (ie: it needs a statement that no MHC molecules found in the vaccines, no evidence for alloreactivity, etc – so this does not seem relevant to the case at hand)

We thank the reviewer for the careful reading and critic.

Lines 241, 247-257, 258, 291-292, 308-311, 319-335: in the revision we have addressed most of the reviewer’s concerns by clarifications and better explanations in the text, by partial rewriting of the discussion, by removing any overt speculation and by adding additional information, discussion and references.

I feel it is written in a way to enhance speculation on the role of contaminants, whereas in fact there seems no plausible link with any side effect (and the most obvious mechanisms – such as contamination with PF4) can be eliminated by the study…I think the manuscript would be greatly enhanced if the authors could:1) to show clear evidence of whether there is a difference in contamination between this and other vaccines (esp JnJ). Eg: sufficient (randomly chosen) batches to show statistical significance (or a clear statement that based on 1 batch versus 3 this is purely anecdotal).

As Above: Choice of lots: All vaccine lots analyzed in this study were obtained in original vials from the Pharmacy of the University Hospital Ulm between March 2021 and September 2021, as they became available in the course of the ongoing vaccination campaign

There has been misunderstanding: a total of 4 AZ (not 3) vaccines and 3 JJ (not 1) vaccines were analysed and all data, including source data, are now presented in the manuscript. This has been explained now in a clearer way in the manuscript.

2) Before speculating on auto-antibodies etc, this should be directly examined (eg: look at antibodies from AZ / JnJ vaccinated and control individuals to cell lysate (western blot? Immunoprecipitation?). These are very easy. Also, make recombinant versions of the top 5 contaminants and see if antibodies to these are detectable in AZ vaccinated vs. control vs. JnJ vaccinated).

Performing additional experiments, for example to screen for the presence of autoantibodies etc. or involving generating recombinant proteins for the top 5 contaminants would have gone far beyond, what reasonably could have been done within a few months of additional experimental work.

3) It should clearly state that although there is a theoretical risk potential contaminants could have an effect, no relevant examples are known in the literature (not my area – but going off reference 29) and no relevant proteins were found. Eg: no PF4 (relevant to HIIT), no MHC (relevant to bovine neonatal pancytopenia).

Lines 241, 247-257, 258, 291-292, 308-311, 319-335: in the revision we have addressed most of the reviewer’s concerns by clarifications and better explanations in the text, by partial rewriting of the discussion, by removing any overt speculation and by adding additional information, discussion and references.

Absence of PF4 and other “high risk” HCPs is mentioned. The relation to bovine neonatal pancytopenia is explained in a better way. Not an immune response to MHC is the issue, rather potential immunogenicity in vaccinees of HEK293-derived HCPs that have polymorphic alleles (resulting in sequence differences at the protein level).